# Vibration Detection and Degraded Image Restoration of Space Camera Based on Correlation Imaging of Rolling-Shutter CMOS

**DOI:** 10.3390/s23135953

**Published:** 2023-06-27

**Authors:** Hailong Liu, Hengyi Lv, Chengshan Han, Yuchen Zhao

**Affiliations:** Changchun Institute of Optics, Fine Mechanics and Physics, Chinese Academy of Sciences, Changchun 130033, China; lv_hengyi@163.com (H.L.); hancs@ciomp.ac.cn (C.H.); zhaoyuchen@ciomp.ac.cn (Y.Z.)

**Keywords:** space camera, rolling-shutter CMOS, correlation imaging, vibration parameter detection, degraded image restoration

## Abstract

To mitigate the influence of satellite platform vibrations on space camera imaging quality, a novel approach is proposed to detect vibration parameters based on correlation imaging of rolling-shutter CMOS. In the meantime, a restoration method to address the image degradation of rolling-shutter CMOS caused by such vibrations is proposed. The vibration parameter detection method utilizes the time-sharing and row-by-row imaging principle of rolling-shutter CMOS to obtain relative offset by comparing two frames of correlation images from continuous imaging. Then, the space camera’s vibration parameters are derived from the fitting curve parameters of the relative offset. According to the detected vibration parameters, the discrete point spread function is obtained, and the rolling-shutter CMOS image degradation caused by vibration is restored row by row. The verification experiments demonstrate that the proposed detection method for two-dimensional vibration achieves a relative accuracy of less than 1% in period detection and less than 2% in amplitude detection. Additionally, the proposed restoration method can enhance the MTF index by over 20%. The experimental results demonstrate that the detection method is capable of detecting high-frequency vibrations through low-frame-frequency image sequences, and it exhibits excellent applicability in both push-scan cameras and staring cameras. The restoration method effectively enhances the evaluation parameters of image quality and yields a remarkable restorative effect on degraded images.

## 1. Introduction

During the operation of a remote sensing satellite in orbit, both internal and external factors can disturb the space attitude of the satellite platform, resulting in vibrations that are transmitted to the space camera [1,2]. This phenomenon results in the relative displacement of the objects’ projection on the focal plane during the integral imaging process of the space camera, thereby compromising image quality [3,4]. With the continuous enhancement in design and manufacturing proficiency in remote sensing optical systems, coupled with the ongoing optimization of performance indices for optical imaging devices, spatial camera resolution is gradually improving. However, these advances have also increased the sensitivity of cameras to vibration, making it an important factor that affects the image quality of high-resolution remote sensors [5]. The detection of vibration parameters is highly significant in the study of satellite platform vibration laws and the enhancement in space camera imaging performance. Additionally, it serves as a fundamental data source for both vibration suppression and degraded image restoration.

Vibration parameter detection methods can be classified into two categories: one involves direct detection using precision acquisition sensors, while the other utilizes digital image processing technology to achieve vibration parameter detection. The direct detection method boasts high sampling frequency and detection accuracy, rendering it a widely adopted technique in space platforms such as Landsat-7, Pleiades, Yaogan-26, and others [6,7,8,9,10]. However, due to spatial constraints, the direct installation of the sensor on the focal plane for vibration parameter measurement is unfeasible. Vibration tests conducted at other locations are attenuated by internal damping measures and the structural conduction of the space camera, resulting in a certain degree of error between collected vibration information and actual focal plane vibration data [11,12]. Early vibration parameter detection methods based on digital image processing are primarily developed for single-frame, fuzzy images of vibrations [13,14]. These methods can only compute the point spread function under specific motion patterns and are susceptible to noise, resulting in reduced accuracy. On the basis of single-frame image detection, a multi-frame image sequence detection method has been developed [15,16,17,18]. In this method, the reference image captured by the high-speed imaging device is utilized to detect vibration parameters through comparison with the main imaging device’s image. However, the reference image is affected by noise due to its short exposure time, which in turn impacts the final detection accuracy. At the same time, according to the sampling theorem, a sampling frequency that is more than twice the vibration frequency is necessary to prevent information loss, so the fast imaging device is required to have a higher frame frequency. For certain space cameras with unique structures, techniques have been developed to utilize parallax observation images for detection, such as the space cameras with overlapping imaging areas [19,20,21,22]. This method necessitates the presence of overlapping pixels in the camera imagery, and the quantity is influenced by the attitude of the satellite.

For images captured by a global-shutter array camera, each pixel is exposed simultaneously, ensuring consistent vibration effects across the entire image. This implies the possibility of utilizing a consistent point spread function (PSF) for image restoration [10,23]. For a rolling-shutter array camera, due to its time-sharing and row-by-row imaging characteristics, there exist differences and coupling between image rows. Therefore, it is impossible to employ a unified point spread function for image restoration [24]. Currently, there is no specialized method for restoring vibration degradation in rolling-shutter array cameras. The only available reference is image restoration technology suitable for linear-array push-scan sensors. Wolberg and Loce formulated the imaging model based on the operational principle of a linear-array push-scan camera, proposed a method for calculating the point spread function of vibration in linear-array cameras, and reconstructed blurred images caused by vibrations [25]. A TDICCD vibration image restoration method was proposed by Zhejiang University which is capable of restoring degraded vibration images row by row through the calculation of point spread functions for each individual row when the instantaneous speed of motion is known [26].

This paper proposes a method for detecting vibration parameters based on correlation imaging of rolling-shutter CMOS. Additionally, a restoration method for degraded images caused by vibration in rolling-shutter CMOS cameras is proposed. These methods are highly applicable to both push-scan and staring cameras. The vibration parameter detection method utilizes the correlation between two consecutive frames of rolling-shutter CMOS images to extract displacement information through comparison. Thereby, the vibration parameters of focal plane position is obtained. Meanwhile, owing to the imaging characteristics of the rolling shutter, the minimum sampling period for detecting vibration parameters is determined by the interval between row exposure commencements. Therefore, the proposed vibration parameter detection method in this paper enables the detection of high-frequency vibration parameters through low-frame-frequency image sequences. According to the detected vibration parameters, the proposed rolling-shutter CMOS point spread function discretization method and image restoration method for vibration degradation can effectively enhance the image quality evaluation index of degraded images.

In this study, a novel vibration detection method is proposed which makes full use of the imaging characteristics of rolling-shutter CMOS and can effectively improve the detection frequency range. At the same time, a novel vibration degradation restoration method for rolling-shutter CMOS is proposed which can effectively improve image quality. In this article, we will introduce the principle and formula derivation of a vibration detection method in the second section, introduce the principle and formula derivation of degraded image restoration method in the third section, introduce the specific experimental method and data results in the fourth section, and present the conclusion in the last section.

## 2. Vibration Parameter Detection

### 2.1. Influence of Vibration

The satellite’s vibration can be decomposed into two components: the vibration along the optical axis and the two-dimensional vibrations perpendicular to each other in a direction orthogonal to the optical axis. Due to the significantly larger object distance compared to the focal length for the space camera, any vibration along the optical axis has negligible impact on image quality. In the directions orthogonal to the optical axis, low-frequency vibrations can cause significant jitter in the camera output, resulting in image position changes that are visually manifested as shifts, stretches, or compressions, and high-frequency vibrations lead to blurred images. Both low-frequency and high-frequency vibrations along the directions orthogonal to the optical axis result in a reduction in the Modulation Transfer Function (MTF) and other image parameters. This paper focuses on the detection of vibration parameters along the directions orthogonal to the optical axis.

### 2.2. Principle of Detection Method

The rolling-shutter CMOS image sensor operates based on the principles of time-sharing and row-by-row imaging. Each row of the sensor sequentially begins and ends its exposure, while each row of images has an equal exposure duration. Therefore, while each row of images is initially independent at the start of exposure, their integration times overlap, resulting in inter-row coupling. Thus, each row of exposure time is staggered and coupled on the temporal axis. Meanwhile, for the image captured by the same row of pixels in two consecutive frames, the time interval at the onset of exposure remains constant and equals that required for capturing one frame of an image. In this paper, we refer to the interrelated imaging method described above as correlation imaging.

In essence, sensor image capture involves the systematic sampling of scene information. The time-sharing and row-by-row imaging of the rolling-shutter CMOS is equivalent to sequentially sampling the scene according to the direction of the shutter. The duration of each sampling corresponds to the exposure time of a single row, while the sampling interval represents the temporal difference between the initial time of two consecutive rows’ exposure. Therefore, each row is subjected to varying magnitudes of vibration-induced impact during the continuous imaging process of a rolling-shutter CMOS. This implies that the corresponding image captures the effects of vibration throughout the time period, while the continuous sequence of images documents all vibrational data during the entire imaging process. The entire array of pixels in the global-shutter sensor is simultaneously exposed during a single time period, resulting in a sampling frequency equivalent to the frame frequency. In contrast, the rolling-shutter sensor significantly enhances the sampling frequency, expands the detection range of vibration frequencies, and enables high-frequency vibration detection through low-frame-rate image sequences.

For analytical purposes, it is assumed that the space camera operates as a staring camera and experiences no relative motion with respect to the rolling-shutter CMOS image sensor, except for vibrations. The relative relationship between two consecutive frames of images is analyzed, as illustrated in Figure 1: the row *i* of the frame *p* and the frame *p* + 1 both capture the same scene with a fixed time interval of $mT_{row}$, where *m* represents the number of rows on the CMOS image sensor and $T_{row}$ denotes the temporal difference between the initial time of two consecutive rows’ exposure, also known as row time. If the imaging is affected by vibration and the period of vibration satisfies condition $T_{vib}\neq mT_{row}/j$, j∈N, a relative offset appears in the scene captured at row *i* of two consecutive frames. However, the data contained in a single row are insufficient for detecting vibration offset. Adjacent rows to row *i* should be utilized for accurate detection. In practical engineering projects, the camera is secured by a flexible device, and the transmitted vibration to the camera is typically below 200 Hz, indicating that the vibration period exceeds 5 ms. While the row time of the rolling-shutter CMOS is on the order of microseconds, it is generally believed that the vibration influences of adjacent rows are consistent, allowing for the expansion of a single-row image into a small image suitable for vibration detection. Image registration methods, such as the gray projection algorithm and normalized cross-correlation algorithm [27,28], can be employed to determine the relative offset between images captured at time $t_{i}$ and time $t_{i}+nT_{row}$ of the same scene by comparing rows *i* − *k* − 1 to *i* + *k* in frame *p* with those in frame *p* + 1.

By segmenting an image frame into multiple blocks by modifying $t_{i}$, the relative offset sequence can be obtained by comparing corresponding blocks between two consecutive frames, and the parameters of the offset curve can be fitted. The sampling interval is determined by the distance between image blocks, whereas the sampling frequency can be adjusted through the modification of this distance.

Within a specific time interval, any vibration can be considered a periodic function that can be decomposed into Fourier series. Therefore, this paper primarily focuses on the study of sinusoidal vibrations. Assuming a sinusoidal function, the vibration equation is expressed as follows:(1)$$y=A_{vib}\sin\left(\omega t+\varphi_{vib}\right)$$
where $\omega =2\pi /T_{vib}$. Within the integration time $T_{e}$, the average offset of row *i* in frame *p*, which is affected by vibration, can be expressed as follows:(2)$${\bar{y}}_{1}=\frac{1}{T_{e}}\int_{t-T_{e}}^{t}A_{vib}\sin\left(\omega t+\varphi_{vib}\right)dt=A_{a}\sin\left(\omega t+\varphi_{a}\right)$$
where $A_{a}=\frac{2A_{vib}}{\omega T_{e}}\sin\left(\frac{\omega T_{e}}{2}\right)$, $\varphi_{a}=\varphi -\frac{\omega T_{e}}{2}$.

In a similar manner, the average offset of row *i* in frame *p* + 1, which is affected by vibration, can be expressed as follows:(3)$${\bar{y}}_{2}=A_{a}\sin\left[\omega \left(t+mT_{row}\right)+\varphi_{a}\right]$$

Then, the relative offset between the row *i* of frame *p* and that of frame *p* + 1 can be expressed as:(4)$$\Delta \bar{y}={\bar{y}}_{1}-{\bar{y}}_{2}=2A_{a}\sin\left(\frac{\omega mT_{row}}{2}\right)\sin\left[\omega \left(t+\frac{mT_{row}}{2}+\frac{T_{vib}}{4}\right)+\varphi_{a}\right]$$

Equation (4) represents the equation for relative offset. Meanwhile, it is assumed that the fitting equation for the detected offset is:(5)$$y=A_{fit}\sin\left(\frac{2\pi }{T_{fit}}t+\varphi_{fit}\right)$$

By comparing Equation (4) with Equation (5), vibration parameters can be obtained from the parameters $A_{fit}$, $T_{fit}$, and $\varphi_{fit}$ of the fitting equation for the detected offset. The specific calculation formula is:(6)$$T_{vib}=T_{fit}$$
(7)$$A_{vib}=\left[A_{fit}/2\sin\left(\frac{\pi mT_{row}}{T_{fit}}\right)\right]\left[\left(\frac{\pi T_{e}}{T_{fit}}\right)/\sin\left(\frac{\pi T_{e}}{T_{fit}}\right)\right]$$
(8)$$\varphi_{vib}=\varphi_{fit}+\frac{\pi T_{e}}{T_{fit}}-\frac{\pi mT_{row}}{T_{fit}}-\frac{\pi }{2}$$

Here, Equation (6) to Equation (8) is the calculation formula of vibration parameters. These equations can be used to calculate both the two-dimensional vibrations perpendicular to each other in a direction orthogonal to the optical axis.

### 2.3. Influence of Image Motion

The preceding section examined the vibration detection of a staring camera, while this section scrutinizes the vibration detection of a push-scan camera and evaluates the impact of image motion on the detection method. For a rolling-shutter CMOS sensor, each row has a consistent exposure time despite starting imaging at different times. Therefore, when the image motion is uniform, the distance of each row’s image motion remains constant, resulting in consistent point spread functions across all rows. It is assumed that the flight direction of the satellite is the *x* axis, the direction perpendicular to the flighting is the *y* axis, and the imaging direction of the rolling shutter is perpendicular to the flight direction. Thus, we need to investigate whether the shift on the *x* axis has any impact on vibration detection. According to the linear system theory, the degraded image g(x,y) under the influence of vibration and image motion is expressed as:(9)$$g\left(x,y\right)=f(x,y)*h(x,y)=\frac{1}{T_{e}}\int_{T_{s}}^{T_{s}+T_{e}}f\left[x-X_{vib}\left(t\right)-X_{img}\left(t\right),y-Y_{vib}\left(t\right)\right]dt$$
where f(x,y) represents the motion-free image and h(x,y) denotes the degradation function whose Fourier spectrum corresponds to the point spread function of motion blur. $X_{vib}(t)$ and $Y_{vib}(t)$ denote the amount of vibration between the camera and object in the *x* and *y* directions at time *t*, while $X_{img}(t)$ represents the degree of image movement in the flight direction.

According to Equation (1), the maximum vibration velocity of sinusoidal function is $v_{vib\max}=2\pi A_{vib}/T_{vib}$. Based on the vibration data obtained from the Yaogan-26 satellite, the maximum vibration velocity can be calculated to be of magnitude $10^{-3}$ m/s. The image motion speed on the focal plane can be estimated to be of magnitude $10^{-1}$ m/s for a space camera with an orbital altitude of 760 km and a focal length of 10 m. Due to the significant disparity between the two, the vibration offset is relatively negligible compared to the image displacement during the exposure time, and it can be considered a constant value throughout the imaging process. Formula (9) can be transformed as follows:(10)$$g\left(x,y\right)=\frac{1}{T_{e}}\int_{T_{s}}^{T_{s}+T_{e}}f\left[x-X_{vib}-X_{img}\left(t\right),y-Y_{vib}\right]dt$$

The Fourier spectrum G(u,v) of the degraded image g(x,y) is represented as follows:(11)$$G\left(u,v\right)=\int_{-\infty }^{+\infty }\int_{-\infty }^{+\infty }g\left(x,y\right)e^{-i2\pi \left(ux+vy\right)}dxdy$$

Substitute Equation (10) into Equation (11) and transform it as follows:(12)$$G\left(u,v\right)=\frac{1}{T_{e}}\int_{T_{s}}^{T_{s}+T_{e}}\left\{\int_{-\infty }^{+\infty }\int_{-\infty }^{+\infty }f\left[x-X_{vib}-X_{img}\left(t\right),y-Y_{vib}\right]e^{-i2\pi \left(ux+vy\right)}dxdy\right\}dt$$

Substitute $x^{\prime }=x-X_{img}\left(t\right)$ and $y^{\prime }=y$ into Equation (12) to obtain:(13)$$G(u,v)=\frac{1}{T_{e}}\int_{T_{s}}^{T_{s}+T_{e}}e^{-i2\pi uX_{img}(t)}[\int_{-\infty }^{+\infty }\int_{-\infty }^{+\infty }f(x^{\prime }-X_{vib},y^{\prime }-Y_{vib})e^{-i2\pi [ux^{\prime }+vy^{\prime }]}dx^{\prime }dy^{\prime }]dt$$

Applying Fourier inverse transform to both sides of Equation (13), we obtain:(14)$$g\left(x,y\right)=\left\{\frac{1}{T_{e}}\int_{-\infty }^{+\infty }\left[\int_{T_{s}}^{T_{s}+T_{e}}e^{-i2\pi uX_{img}\left(t\right)}dt\right]e^{i2\pi ux}du\right\}*f\left(x-X_{vib},y-Y_{vib}\right)$$

Take $PSF_{img}\left(x\right)=\frac{1}{T_{e}}\int_{-\infty }^{+\infty }\left[\int_{T_{s}}^{T_{s}+T_{e}}e^{-i2\pi uX_{img}\left(t\right)}dt\right]e^{i2\pi ux}du$; then, Equation (14) can be transformed as follows:(15)$$g\left(x,y\right)=PSF_{img}\left(x\right)*f\left(x-X_{vib},y-Y_{vib}\right)$$

Meanwhile, $PSF_{img}\left(x\right)$ can be transformed as follows:(16)$$PSF_{img}\left(x\right)=\frac{1}{T_{e}}\int_{T_{s}}^{T_{s}+T_{e}}\left\{\int_{-\infty }^{+\infty }e^{i2\pi \left[ux-uX_{img}\left(t\right)\right]}du\right\}dt=\frac{1}{T_{e}}\int_{T_{s}}^{T_{s}+T_{e}}\delta \left[x-X_{img}\left(t\right)\right]dt$$

Due to $V_{img}\left(t\right)=dX_{img}\left(t\right)/dt$, Equation (16) is transformed into:(17)$$PSF_{img}\left(x\right)=\frac{1}{T_{e}}\int_{X_{img\min}}^{X_{img\max}}\frac{\delta \left[x-X_{img}\left(t\right)\right]}{V_{img}\left(t\right)}dX_{img}\left(t\right)$$

Image motion is characterized by uniform motion, which can be mathematically expressed as $V_{img}\left(t\right)=V_{img}$ and subsequently substituted into Equation (17):(18)$$PSF_{img}\left(x\right)=\frac{1}{T_{e}V_{img}},\;x\in \left[0,T_{e}V_{img}\right]$$

According to Equation (18), function $PSF_{img}\left(x\right)$ is a normalized function.

The vibration offset of a specific image row, as shown in Equation (15), is the result of synthesizing multiple rows involved in $PSF_{img}\left(x\right)$. In other words, the offset ${\bar{y}}^{\prime }$ of an image affected by vibration under the influence of image motion can be expressed as:(19)$${\bar{y}}^{\prime }=PSF_{img}\left(x\right)*y\left(x\right)$$

Moreover, as the vibration offset remains constant over a period of time $T_{e}$, where y(x) is a fixed value and $y\left(x\right)=\bar{y}$, Equation (19) can be transformed as follows:(20)$${\bar{y}}^{\prime }=\bar{y}\sum PSF_{img}\left(x\right)$$

As $PSF_{img}\left(x\right)$ serves as a normalization function, Equation (20) is thereby transformed as follows:(21)$${\bar{y}}^{\prime }=\bar{y}$$

According to Equation (21), if the velocity of image motion greatly exceeds that of the vibration, the offset caused by vibration in the image will not be affected by the image motion. Therefore, the presence of image motion does not significantly affect the correlation vibration parameter detection method of rolling-shutter CMOS. Even in the presence of image motion, vibration parameters can still be directly detected without taking its influence into consideration. In conclusion, the proposed method in this paper is universally applicable to various types of space cameras, including push-scan and staring cameras, exhibiting extensive versatility.

## 3. Degraded Image Restoration

After obtaining the vibration parameters through the proposed vibration detection method in this paper, image restoration of degraded images can be achieved. However, due to the time-sharing and row-by-row exposure imaging characteristics of rolling-shutter CMOS, it differs from global-shutter CMOS. The subsequent sections present a comprehensive introduction to the discretization method of point spread function and the image restoration method for vibration degradation.

### 3.1. Method of PSF Discretization

In order to facilitate the calculation, the influence of noise and image motion is not considered in the derivation of the formula. Then, the degraded image g(x,y) can be obtained from Equation (9) as follows:(22)$$g\left(x,y\right)=f(x,y)*h(x,y)=\frac{1}{T_{e}}\int_{T_{s}}^{T_{s}+T_{e}}f\left[x-X_{vib}\left(t\right),y-Y_{vib}\left(t\right)\right]dt$$

Equation (22) is a continuous function; however, digital images are discrete in nature. Therefore, prior to any digital image processing, discretization must be performed. For rolling-shutter CMOS, the exposure time $T_{e}$ is an integral multiple ($N_{e}$) of the line time, namely, $T_{e}=N_{e}T_{row}$, which can be obtained by substituting into Equation (22):(23)$$g\left(x,y\right)=\frac{1}{N_{e}T_{row}}\sum_{i=1}^{N_{e}}\int_{T_{s}+\left(i-1\right)T_{row}}^{T_{s}+iT_{row}}f\left[x-X_{vib}\left(t\right),y-Y_{vib}\left(t\right)\right]dt$$

For the vibration with a vibration function of $y=A_{vib}\sin\left(\omega t+\varphi_{vib}\right)$, the maximum distance of vibration within $T_{row}$ is:(24)$$\Delta s_{max}=2A_{vib}\sin\left(\pi T_{row}/T_{vib}\right)$$

Taking the vibration parameters of the Yaogan-26 satellite as an example, when substituted into Equation (24), the resulting vibration distance within time $T_{row}$ will not exceed 0.1 pixel. In the range $\left[T_{s}+\left(i-1\right)T_{row},T_{s}+iT_{row}\right]$, $f\left[x-X_{vib}\left(t\right),y-Y_{vib}\left(t\right)\right]$ can be regarded as a constant value.

If we define that $f_{i}=f\left\{x-X_{vib}\left[T_{s}+\left(i-0.5\right)T_{row}\right],y-Y_{vib}\left[T_{s}+\left(i-0.5\right)T_{row}\right]\right\}$, then Equation (23) can be transformed as follows:(25)$$g\left(x,y\right)=\frac{1}{N_{e}}\sum_{i=1}^{N_{e}}f_{i}$$

Define $X_{i}=X_{vib}\left[T_{s}+\left(i-0.5\right)T_{row}\right]$ and $Y_{i}=Y_{vib}\left[T_{s}+\left(i-0.5\right)T_{row}\right]$. Since $X_{i}$ and $Y_{i}$ may not be integers, and pixel coordinates in digital images must be integer values, the linear interpolation method is employed for fitting. Define $X_{i\min}=\left|X_{i}\right|$ and $Y_{i\min}=\left|Y_{i}\right|$, where | ⋅ | represents rounding down to the nearest integer, and define:(26)$$f_{i11}=f(x-X_{i\min},y-Y_{i\min})$$
(27)$$f_{i12}=f(x-X_{i\min},y-Y_{i\min}-1)$$
(28)$$f_{i21}=f(x-X_{i\min}-1,y-Y_{i\min})$$
(29)$$f_{i22}=f(x-X_{i\min}-1,y-Y_{i\min}-1)$$

By utilizing linear interpolation fitting, we can obtain:(30)$$\begin{array}{l} f_{i}=f_{i11}(X_{i\min}+1-X_{i})(Y_{i\min}+1-Y_{i})+f_{i12}(X_{i\min}+1-X_{i})(Y_{i}-Y_{i\min})\\ \;\;\;\;+f_{i21}(X_{i}-X_{i\min})(Y_{i\min}+1-Y_{i})+f_{i22}(X_{i}-X_{i\min})(Y_{i}-Y_{i\min}) \end{array}$$

Here it is defined:
(31)$$PSF_{i}(x-X_{i\min},y-Y_{i\min})=(X_{i\min}+1-X_{i})(Y_{i\min}+1-Y_{i})$$
(32)$$PSF_{i}(x-X_{i\min},y-Y_{i\min}-1)=(X_{i\min}+1-X_{i})(Y_{i}-Y_{i\min})$$
(33)$$PSF_{i}(x-X_{i\min}-1,y-Y_{i\min})=(X_{i}-X_{i\min})(Y_{i\min}+1-Y_{i})$$
(34)$$PSF_{i}(x-X_{i\min}-1,y-Y_{i\min}-1)=(X_{i}-X_{i\min})(Y_{i}-Y_{i\min})$$

If Equations (31) to (34) are substituted into Equation (30), the following results can be obtained:(35)$$\begin{array}{l} f_{i}=f_{i11}PSF_{i}(x-X_{i\min},y-Y_{i\min})+f_{i12}PSF_{i}(x-X_{i\min},y-Y_{i\min}-1)\\ \;\;\;\;+f_{i21}PSF_{i}(x-X_{i\min}-1,y-Y_{i\min})+f_{i22}PSF_{i}(x-X_{i\min}-1,y-Y_{i\min}-1) \end{array}$$

Equation (35) is supplemented and complete within the whole image range, that is, except for PSFS at the above four pixel positions; all other PSFs are 0. Then, Equation (35) can be transformed as follows:(36)$$f_{i}=\sum_{x,y}f(x,y)\cdot PSF_{i}(x,y)$$
where $PSF_{i}(x,y)$ is:(37)$$PSF_{i}(x,y)=\left[\begin{array}{cccccccc} 0 & \cdots & 0 & 0 & 0 & 0 & \cdots & 0\\ \vdots & \ddots & \vdots & \vdots & \vdots & \vdots & ⋰ & \vdots \\ 0 & \cdots & 0 & 0 & 0 & 0 & \cdots & 0\\ 0 & \cdots & 0 & PSF_{i21} & PSF_{i11} & 0 & \cdots & 0\\ 0 & \cdots & 0 & PSF_{i22} & PSF_{i12} & 0 & \cdots & 0\\ 0 & \cdots & 0 & 0 & 0 & 0 & \cdots & 0\\ \vdots & ⋰ & \vdots & \vdots & \vdots & \vdots & \ddots & \vdots \\ 0 & \cdots & 0 & 0 & 0 & 0 & \cdots & 0 \end{array}\right]$$

Substituting Equation (36) into Equation (25), we obtain:(38)$$g(x,y)=\frac{1}{N_{e}}\sum_{i=1}^{N_{e}}\sum_{x,y}f(x,y)\cdot PSF_{i}(x,y)=\sum_{x,y}\left\{f(x,y\right)\cdot [\frac{1}{N_{e}}\sum_{i=1}^{N_{e}}PSF_{i}(x,y)]\}$$

In contrast to Equation (22), it is evident that:(39)$$PSF(x,y)=\frac{1}{N_{st}}\sum_{i=1}^{N_{st}}PSF_{i}(x,y)$$

Based on known parameters of vibration, the discrete point spread function of vibration can be derived utilizing Equation (39).

### 3.2. Degraded Image Restoration of Rolling-Shutter CMOS

In the rolling-shutter CMOS camera, the exposure start time of each row varies, resulting in a unique point spread function for each row. Therefore, the utilization of a unified PSF for complete image restoration is unfeasible, necessitating row-by-row restoration.

Figure 2 illustrates the operation of a rolling-shutter CMOS camera: the pixel located in row *k* performs exposure imaging within the range of $T_{k}$ to $T_{k}+N_{e}T_{row}$, and undergoes a focal plane displacement of $D_{k}$ during the exposure process; in a similar manner, the pixel located in row *k* − *l* performs exposure imaging within the range of $T_{k-l}$ to $T_{k-l}+N_{e}T_{row}$, and undergoes a focal plane displacement of $D_{k-l}$ during the exposure process; the pixel located in row *k* + *l* performs exposure imaging within the range of $T_{k+l}$ to $T_{k+l}+N_{e}T_{row}$, and undergoes a focal plane displacement of $D_{k+l}$ during the exposure process. It is evident that the vibration distance varies during the imaging process of different rows, indicating a distinct PSF for each row. However, the vibration effects overlap in several adjacent rows. By appropriately selecting l, we can approximate that the vibration effect on the image remains consistent from row *k* − *l* to row *k* + *l*. The vibration effect of these rows is consistent with that of row *k*, indicating that the point spread functions within this range are identical to those of row *k*. Thus, the point spread function of row *k* can be utilized for image restoration of rows *k* − *l* to *k* + *l*. The central row of the restored image is then designated as the rolling-shutter CMOS restored image’s row *k*. Following this process, the same operation is applied to each row of the image, resulting in a restored rolling-shutter CMOS image of the entire frame.

## 4. Experiment

### 4.1. Platform of Vibration Detection

The experimental equipment mainly included an optical platform, two-dimensional vibration device, signal generator, rolling-shutter CMOS camera, scene target, etc. The experimental platform is shown in Figure 3.

### 4.2. Experiment of Vibration Detection and Result Analysis

Since the motion is relative, the target motion was used in the experiment to simulate camera vibration. A two-dimensional vibration device was used to add vibration excitation to the target in both horizontal and vertical directions. The equivalent vibration parameters on the focal plane after conversion are shown in Table 1. The direction of the camera rolling shutter was horizontal. First, we selected the ISO 12233 camera resolution chart as the scene target. The rolling-shutter CMOS camera has a row time of 20.52 μs, with an exposure time that is 20 times the row time (410.4 μs). The correlation images captured in two consecutive frames are presented in Figure 4. It can be seen from Figure 4a,b that under the influence of vibration, the straight line in the horizontal direction distorted, while the straight line in the vertical direction shifted in the transversal direction. There were also some differences in vibration effects between frame *p* and frame *p* + 1.

The two frames of continuously exposed images were processed by windowing, and the appropriate windowing size and windowing interval were selected. The windowing interval was the sampling frequency of the vibration detection. The two groups of windowing image sequences are shown in Figure 5. By comparing the image sequence, we can observe that there are displacements in both the horizontal and vertical directions between the two frames, indicating that they were affected by vibration, resulting in twisting and stretching. Furthermore, there are dissimilarities between the impacts on frame *p* and frame *p* + 1.

A vibration parameter detection algorithm based on rolling-shutter CMOS correlation imaging was utilized to compute the gray projection of corresponding image pairs in two sets of windowing image sequences. The relative offset array of vibration between two continuous exposure rolling-shutter CMOS correlation images was subsequently obtained. By performing Fourier fitting on the array, we can obtain the fitting curve and its corresponding parameters for vibration offset. The offset fitting curves for the horizontal and vertical directions are depicted in Figure 6.

According to the fitting curve parameters, the horizontal and vertical vibration parameters can be obtained from Equations (6) to (8), and the results are shown in Table 2.

In order to simulate the in-orbit imaging state, the scene target was replaced by a satellite remote sensing image printed by the HD printer. The two frames of correlation images of continuous exposure are shown in Figure 7.

After the same process, the results are shown in Table 3.

From Table 2 and Table 3, the results demonstrate that the proposed vibration parameter detection method in this paper is capable of effectively detecting the vibration parameters of space cameras with high accuracy, with a relative accuracy for period detection less than 1% and amplitude detection less than 2%. The detection of 50 Hz vibration requires a global-shutter imaging device to operate at a minimum frame rate of 100 fps. However, the rolling-shutter CMOS requires only 24 fps to accurately detect high-frequency vibrations, thereby achieving the objective of detecting such vibrations through a low-frame-rate image sequence.

### 4.3. Experiment of Image Restoration and Result Analysis

According to the proposed image restoration method in this paper, a degraded image can be restored by utilizing detected vibration parameters. The pre- and post-restoration comparison of ISO 12233 as the scene target is illustrated in Figure 8, while the pre- and post-restoration comparison of a satellite remote sensing image as the scene target is depicted in Figure 9.

The visual improvement in the twisting and stretching of the image after restoration is readily apparent. Subsequently, the evaluation parameters for correlation images pre- and post-restoration were computed to assess the effectiveness of image restoration based on the resulting data. The imaging quality evaluation parameters used in this paper include: Peak Signal-to-Noise Ratio (PSNR), Structural Similarity Index Measurement, SSIM, and MTF. PSNR is a metric used to quantify the level of signal distortion between images; higher values indicate lower levels of distortion. SSIM is a metric utilized for gauging the likeness between two images; values range from 0 to 1. The closer the value is to 1, the higher the degree of similarity; conversely, as it approaches 0, so does the level of resemblance. MTF is a crucial parameter for the objective assessment of camera imaging quality, as it reflects the camera’s ability to transmit frequency domain information. In this paper, MTF is determined using the slanted-edge method, and subsequently, the MTF values for both horizontal and vertical directions are computed separately. Imaging quality evaluation parameters pre- and post-restoration are presented in Table 4.

The experimental results demonstrate that the proposed degraded image restoration method can significantly enhance the evaluation parameters of imaging quality. In particular, the proposed method exhibits a significant improvement in MTF by over 20%, indicating its efficacy in mitigating image quality degradation caused by vibration. Moreover, this indirectly validates the precision of the vibration parameter detection method.

## 5. Conclusions

A vibration parameter detection method based on correlation imaging of rolling-shutter CMOS is proposed in this paper. The method can effectively and accurately detect the vibration parameters of space cameras. The period detection has a relative accuracy of less than 1%, while the amplitude detection has a relative accuracy of less than 2%. Furthermore, the utilization of rolling-shutter CMOS time-sharing and row-by-row imaging principles enables the detection of high-frequency vibration through a low-frame-frequency image sequence. After detecting the vibration parameters and obtaining the discrete point spread function, the proposed method of degraded image restoration for rolling-shutter CMOS can effectively enhance image quality evaluation parameters. In particular, the proposed method demonstrates a significant improvement of over 20% in MTF, indicating its effectiveness in restoring image degradation caused by vibration. The methods proposed in this paper are universally applicable to various types of space cameras, including push-scan and staring cameras. These methods possess broad applicability and hold significant implications for suppressing vibrations in space cameras.

## Figures and Tables

**Figure 1 sensors-23-05953-f001:**
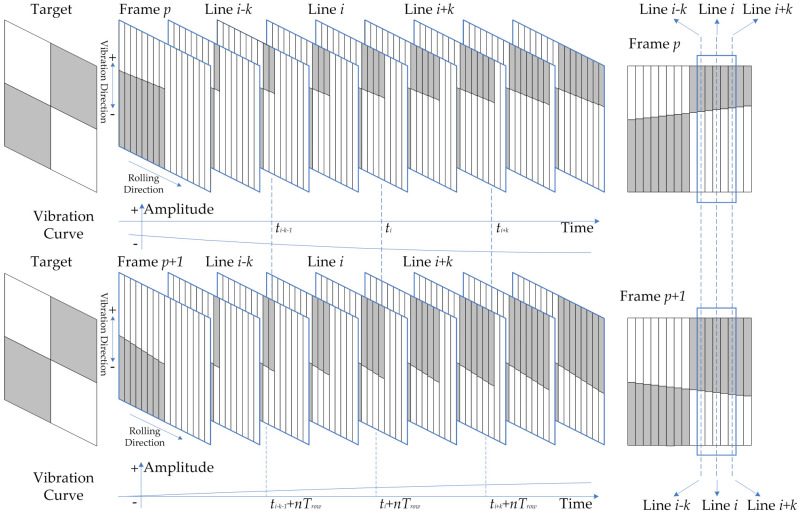
Correlation imaging of rolling-shutter CMOS.

**Figure 2 sensors-23-05953-f002:**
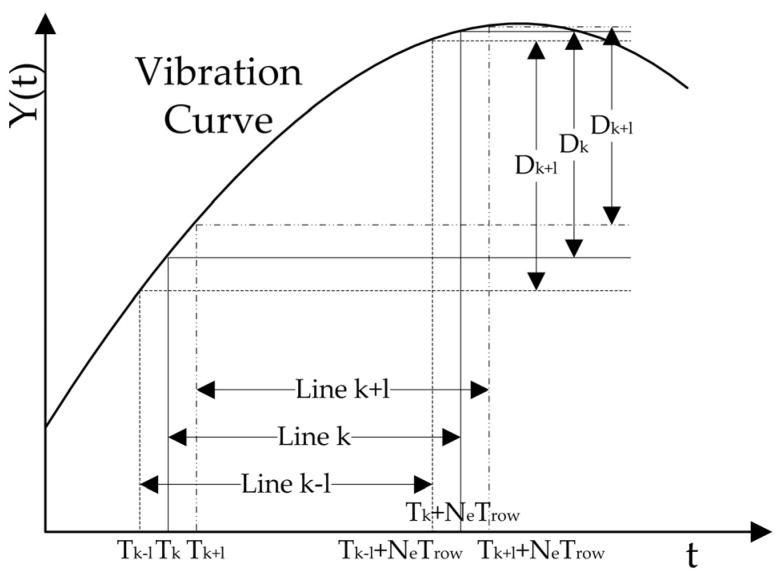
The impact of vibration on various rows of rolling-shutter CMOS.

**Figure 3 sensors-23-05953-f003:**
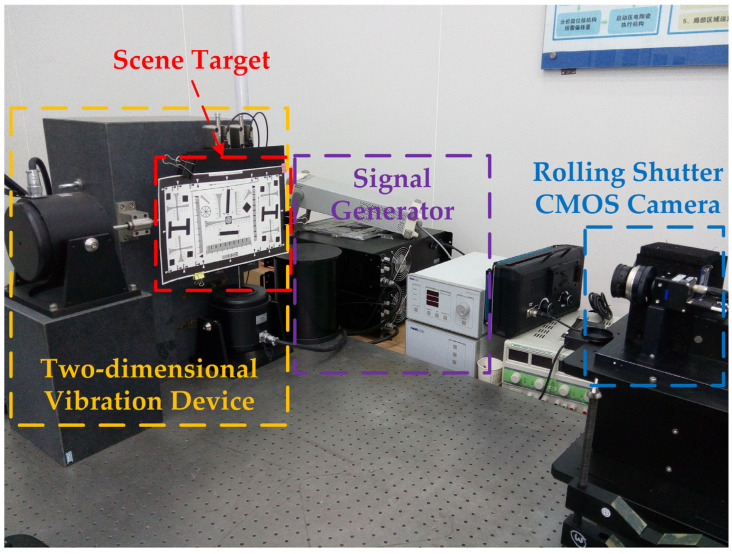
Platform of vibration detection.

**Figure 4 sensors-23-05953-f004:**
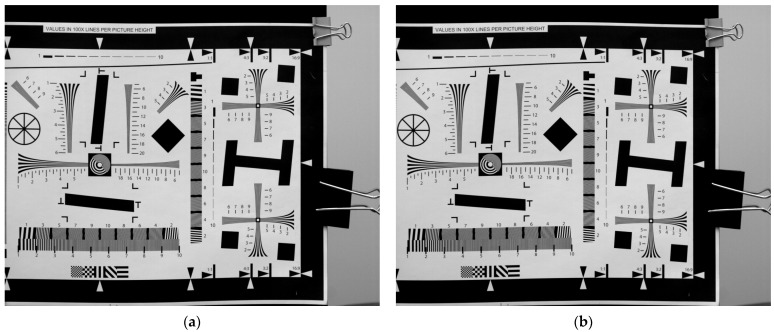
Correlation image sequence impacted by vibration (ISO 12233). (**a**) Frame *p*; (**b**) frame *p* + 1.

**Figure 5 sensors-23-05953-f005:**
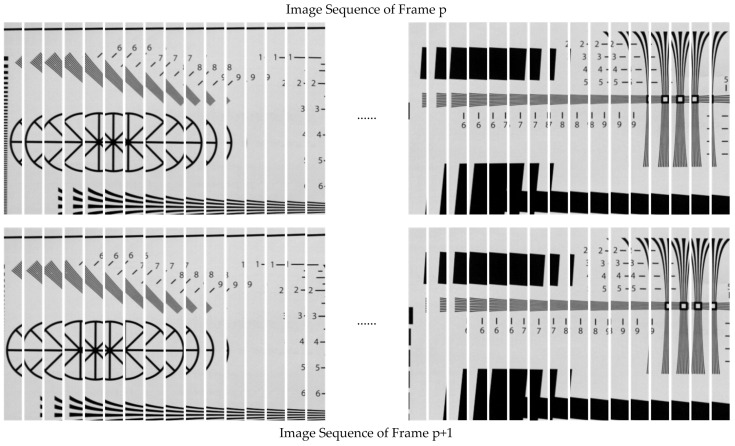
Windowing of correlation image.

**Figure 6 sensors-23-05953-f006:**
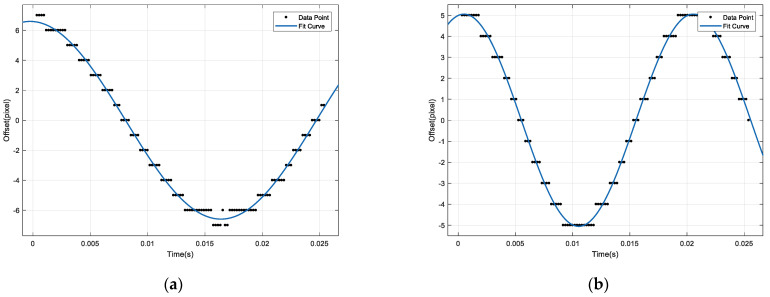
Relative offset fitting curve. (**a**) Horizontal; (**b**) vertical.

**Figure 7 sensors-23-05953-f007:**
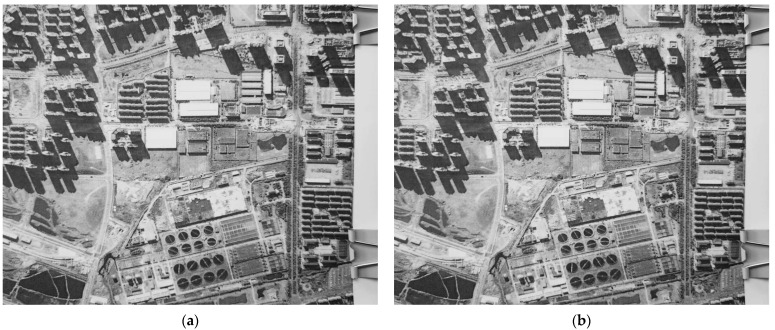
Correlation image sequence impacted by vibration (in-orbit simulation). (**a**) Frame *p*; (**b**) frame *p* + 1.

**Figure 8 sensors-23-05953-f008:**
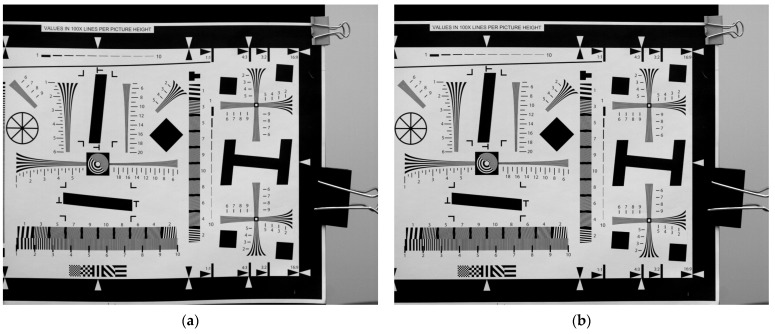
Pre- and post-restoration comparison of ISO 12233 as the scene target. (**a**) Pre-restoration; (**b**) post-restoration.

**Figure 9 sensors-23-05953-f009:**
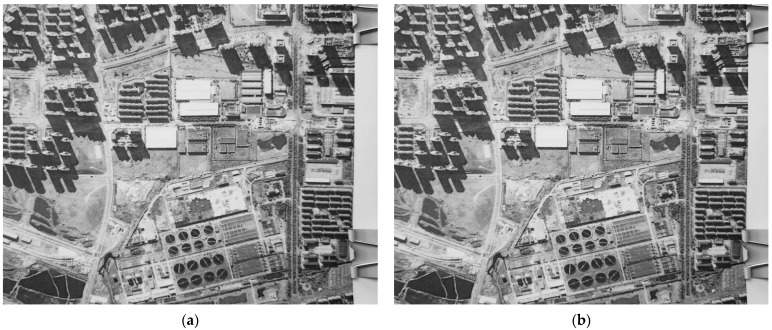
Pre- and post-restoration comparison of in-orbit simulation as the scene target. (**a**) Pre-restoration; (**b**) post-restoration.

**Table 1 sensors-23-05953-t001:** The equivalent vibration parameters of the focal plane.

Vibration Direction	Frequency (Hz)	Period (ms)	Amplitude (pixel)
Horizontal	50	20	3.1
Vertical	30	33.33	5.2

**Table 2 sensors-23-05953-t002:** Vibration detection results and analysis (ISO 12233).

Vibration Direction	Input Vibration Parameters		Detected Vibration Parameters			Result of Contrast	
	Period (ms)	Amplitude (pixel)	Period (ms)	Amplitude (pixel)	Phase (π)	Relative Accuracy of Period (%)	Relative Accuracy of Amplitude (%)
Horizontal	20	3.1	20.048	3.147	0.653	0.242	1.526
Vertical	33.33	5.2	33.280	5.195	1.234	0.161	0.096

**Table 3 sensors-23-05953-t003:** Vibration detection results and analysis (in-orbit simulation).

Vibration Direction	Input Vibration Parameters		Detected Vibration Parameters			Result of Contrast	
	Period (ms)	Amplitude (pixel)	Period (ms)	Amplitude (pixel)	Phase (π)	Relative Accuracy of Period (%)	Relative Accuracy of Amplitude (%)
Horizontal	20	3.1	19.890	3.084	0.638	0.519	0.551
Vertical	33.33	5.2	33.122	5.211	1.227	0.218	0.635

**Table 4 sensors-23-05953-t004:** Imaging quality evaluation parameters pre- and post-restoration.

Scene Target Type	Evaluation Parameters	Degraded Image		Restored Image		Improved Effect (%)	
		Frame *p*	Frame *p* + 1	Frame *p*	Frame *p* + 1	Frame *p*	Frame *p* + 1
ISO 12233	PSNR	33.915	34.068	40.525	40.817	19.49	19.81
	SSIM	0.9286	0.9338	0.9971	0.9978	7.37	6.85
	MTF(Horizontal)	0.0459	0.1472	0.0805	0.1779	75.38	20.86
	MTF(Vertical)	0.1035	0.1009	0.1477	0.1443	42.71	43.01
In-orbit simulation	PSNR	29.563	29.602	38.324	38.682	29.64	30.68
	SSIM	0.8782	0.8868	0.9963	0.9965	13.45	12.37

## Data Availability

No new data were created or analyzed in this study. Data sharing does not apply to this article.

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
