# Peer review of "Vibration Detection and Degraded Image Restoration of Space Camera Based on Correlation Imaging of Rolling-Shutter CMOS"

_sensors, 2023, doi:10.3390/s23135953_

Round 1

Reviewer 1 Report

In this paper, a vibration parameter detection method based on autocorrelation imaging of the rolling shutter CMOS is proposed, which can effectively and accurately detect the vibration parameters of space cameras. It has been found that the use of rolling shutter CMOS timesharing and row-by-row imaging principles enables the detection of high frequency vibration through a low frame frequency image sequence. The paper is timely and technically sound. On the other hand, it should be reviewed in terms of organization and presentation. My evaluations, concerns, and suggestions on the paper are listed below.

·         Some sentences are very long, especially in the Abstract and Introduction Sections. Plain expression should be used in a way to preserve the integrity of meaning.

·         Subject presentation matter should be improved.

·         At the end of the introduction section, there should be a paragraph about the novelty of the proposed and performed work (Ex: In this paper or study ….).

·         The last paragraph of the Introduction section should mention the organization of the rest of the article.

·         Since there is a large number of similar studies in the literature (as listed below), the the innovative aspect of the study and its difference from similar studies should be emphasized in a comparative manner.

o   He, L., Cui, G., Feng, H., Xu, Z., Li, Q., & Chen, Y. (2015). Fast image restoration method based on coded exposure and vibration detection. Optical Engineering54(10), 103107-103107.

o   Yue, R., Wang, H., Jin, T., Gao, Y., Sun, X., Yan, T., ... & Wang, S. (2021). Image motion measurement and image restoration system based on an inertial reference laser. Sensors21(10), 3309.

o   Changyoon Yi, Jaewoo Jung, Jeongmyo Im, Kyung Chul Lee, Euiheon Chung, and Seung Ah Lee, "Single-shot temporal speckle correlation imaging using rolling shutter image sensors," Optica 9, 1227-1237 (2022)

·         Abstract and Conclusion Sections should be improved. The results obtained for the method used, should be included in the Abstract and Conclusion sections.

·         PSNR and SSIM performance of the proposed method in realistic satellite images should be examined.

·         There are a few grammatical mistakes and typos. Authors need to read carefully and fix the issues.

o   Leave a space between the reference and the previous word. For example, 2nd page, line 51 “vibration data[10,11].”, line 53, line 56 … etc.

o   5th page, 184th line: Authors state that, “Thus, We need...” Capital letters should be used instead of uppercase letters.

Minor editing of English language required

Reviewer 2 Report

Several terms used throughout the paper are not the conventional ones and this must to be addressed. For instance, MTF is the "Modulation Transfer Function" not "Module Transfer Function". Autocorrelation has a clear meaning and its a mathematic function of extrem importance and clearly defined. I dont think autocorrelation imaging is a good term to refer to the the "principles" of time-sharing and row-by-row imaging of the so called rolling shutter CMOS image sensor.

The concept of "error" is a difficult one in metrology and the use of this term should be avoid either if it refers to the difference bettween the measured value and a "true" (assumed as such... because it does not exists! Any value, or measured value, of an "entity" have associated an uncertainety) value, or if it refers, wrongly, to the "uncertainty" of a measure.

In spite these metrological problems that should be dealt with, the method proposed is simple enought to be robust  and therefore should be considered as a useful way to reduced at a certain extent the effects of transversal vibrations (maybe the term "vertical" is not the best one, on sake of generalization)

Although acceptable a revision the English writing will improve paper' readability

Reviewer 3 Report

Report of the manuscript “Vibration detection and degraded image restoration of space camera based on autocorrelation imaging of rolling shutter CMOS”.

The manuscript under review seemed very interesting and actual to the reviewer. However, the reviewer believes that the manuscript contains significant shortcomings and therefore needs major revision.

Major comment

1.

In subsection 2.1 (lines 99, 100), the authors pay attention to “two-dimensional vibration perpendicular to each other in a direction orthogonal to the optical axis.” The reviewer agrees with this statement.

However, in lines 100-106, the authors state that “In the direction of the vertical optical axis, low-frequency vibrations can cause significant jitter in the camera output, …” The authors further narrow the purpose of the article: “This paper focuses on the detection of vibration parameters along the vertical optical axis direction.” (lines 107-108). The reviewer does not agree with the authors’ arguments and believes that the authors should take into account vibration in both vertical and horizontal directions. The authors should add to subsection 2.2 a description of the vibration in the horizontal direction.

2.

The authors should take into account the motion of the satellite relative to the surface of the Earth, even if it is a geostationary satellite.

3.

What is the difference between 6 images: Fig. 4 a and b, Fig. 7 a and b, and Fig. 8 a and b? I could not solve this puzzle. These images are absolutely identical, in my opinion. The authors should clarify these differences in the text. Perhaps the authors will add insets to the photos so that the difference is visible.

Technical corrections

4.

Change “the objects projection” to “the objects’ projection” (line 32)

5. 

Change “George Wolberg” to “Wolberg and Loce” (line 75)

Minor editing of the English language is required.

Round 2

Reviewer 1 Report

The authors have progressed in improving the paper compared to previous versions of the paper (sensors-2434790-peer-review-v1 & sensors-2434790-peer-review-v2). When the previous and revised versions of the paper are evaluated together, it is seen that the authors make the corrections requested by the referees and show the necessary sensitivity in the revision of the paper in line with the comments. In the revised version of the paper, all the comments have been considered and addressed by the authors.

The response to reviewers file is well-prepared. The changes made by the authors in line with the opinions/suggestions/evaluations of the referees can be tracked. The existing organization and spelling problems in the previous version of the article have been fixed. In the revised version, the clarity and follow-up of the study have been increased. In addition, the article has been carefully reviewed for grammatical and typos.

As a result, my concerns on the previous version of the paper have disappeared with the explanations made by the authors, as well as the revision they have made.

This revision is sufficient, and it is possible to evaluate the paper for publication after preparation according to MDPI template.

Reviewer 3 Report

The authors consistently answered all my comments and suggestions.

I can recommend the manuscript for publication.

The manuscript requires minor editing of the English language.

All sentences and phases are written clearly so that their meaning is not in doubt.

However, some phrases need to be corrected.

Example of such a phrase (lines 82-83)

“As well as a restoration method for degraded images caused by vibration in rolling shutter CMOS cameras is proposed.”